Biodiversity surveys reveal eight new species of freshwater crabs (Decapoda: Brachyura: Potamidae) from Yunnan Province, China

http://orcid.org/0000-0001-9212-5246 Naruse Tohru 1
Chia Jing En 2
Zhou Xianmin 3 4 zhouxmjxmu@ncu.edu.cn
1 Tropical Biosphere Research Center, Iriomote Station, University of the Ryukyus , Okinawa , Japan
2 Department of Biological Sciences, Faculty of Science, National University of Singapore , Singapore , Republic of Singapore
3 Research Lab of Freshwater Crustacean Decapoda & Paragonimus, School of Basic Medical Sciences, Nanchang University , Nanchang , People’s Republic of China
4 Key Laboratory of Poyang Lake Environment and Resource Utilization, Ministry of Education, Nanchang University , Nanchang , People’s Republic of China
Gao Junkuo
Electronic publication date: 2018 Sep 7
Publication date: 2018
Volume: 6
Electronic Location ID: e5497
Received 2018 Jan 2; Accepted 2018 Jul 31
Copyright: © 2018 Naruse et al.
Copyright year: 2018
Copyright holder: Naruse et al.
License: This is an open access article distributed under the terms of the Creative Commons Attribution License, which permits unrestricted use, distribution, reproduction and adaptation in any medium and for any purpose provided that it is properly attributed. For attribution, the original author(s), title, publication source (PeerJ) and either DOI or URL of the article must be cited.
License URL: https://creativecommons.org/licenses/by/4.0/

Keywords: Indochinamon, Parvuspotamon, Pararanguna, Potamiscus, Taxonomy

Funding: National University of Singapore National Science Foundation of China 81260257, 31460156, 31560179 Natural Science Foundation of Jiangxi, China 20171BAB214013 Innovation Special Foundation for Graduates of Nanchang University CX2016280 National Sharing Service Platform for Parasite Resources TDRC-22 A research grant on Indochinese arthropods was received from the National University of Singapore. This work is supported by the National Science Foundation of China (Grant No. 81260257, 31460156, 31560179), the Natural Science Foundation of Jiangxi, China (Grant No. 20171BAB214013), the Innovation Special Foundation for Graduates of Nanchang University (Grant No. CX2016280) and the National Sharing Service Platform for Parasite Resources (TDRC-22). The funders had no role in study design, data collection and analysis, decision to publish, or preparation of the manuscript.

==============================
Yunnan Province is known to host the highest species diversity of the true freshwater crabs in China; 50 species have been recorded from the province by 2017. In 2004, our team conducted a biodiversity survey of the freshwater crabs in Yunnan Province to determine how well the diversity of crabs in the area has been characterized. We collected a total of 25 species, of which nine species proved to be new to science, and eight of which are described here. These include four species of the genus Indochinamon Yeo & Ng, 2007, two species of the genus Potamiscus Alcock, 1909, and one species each of the genera Pararanguna Dai & Chen, 1985, and Parvuspotamon Dai & Bo, 1994. The new species of Pararanguna and Parvuspotamon represent the second species of respective genera, which are here redefined. Detailed comparisons with morphologically allied species are provided. Photographs of the type specimens of their comparative species which are poorly illustrated in the literature are also provided to allow better understanding of their morphology. This study brings the number of the freshwater crabs of Yunnan Province to 58. Since about 13.8% of the number of species (eight out of 58 species) is increased by surveys conducted within a relatively short period, it is most probable that the species diversity of this group is still understudied in Yunnan Province.

Introduction

China is the world’s most species-rich country for the freshwater crabs, with more than 200 species recorded from the country (Dai, 1999; Cumberlidge et al., 2009, 2011; Cumberlidge & Ng, 2009). Cumberlidge et al. (2011: 46) had made the observation that “(d)ozens of new species remain undescribed,” and several recent discoveries have proven this to be true (Cheng, Lin & Li, 2010; Zhu, Naruse & Zhou, 2010; Huang, Huang & Ng, 2012; Lin, Cheng & Chen, 2012, 2013; Naruse, Zhu & Zhou, 2013; Huang, Mao & Huang, 2014; Do, Shih & Huang, 2016; Huang, Shih & Mao, 2016; Chu, Sun & Sun, 2017; Chu, Zhou & Sun, 2017; Huang, Ahyong & Shih, 2017; Huang, Shih & Ng, 2017; Ng, 2017; Huang, 2018).

Yunnan Province (Fig. 1) is located in southwestern China at the meeting point of the eastern Asia monsoon region, the Tibetan Plateau region and the tropical monsoon region of southern Asia and Indo-China. The significantly wide elevation range within the province (76.4–6,740 m) and the presence of at least six major river systems (Irrawaddy, Salween, Mekong, Yangtze, Red, and Pearl) have contributed a wide range of habitats, topographies, and its rich biota (Kunming Institute of Zoology, CAS, 1999; Yang et al., 2004). Indeed, Yunnan hosts 50 species of the freshwater crab in 15 genera (Dai, 1999; Dai & Cai, 1998; Naruse, Yeo & Zhou, 2008; Chu, Zhou & Sun, 2017), which represents the highest number of the freshwater crab species in the provinces of China.

Figure 1 Map showing Yunnan Province, China, and collection localities of seven new species described in this study.

Ia, Indochinamon ahkense sp. n.; Ip, I. parpidum sp. n.; It, I. tujiense sp. n.; Il, I. lui sp. n.; Prh, Pararanguna hemicyclia sp. n.; Pvd, Parvuspotamon dixuense sp. n.; Pof, Potamiscus fumariatus sp. n.

In 2004, a team from the National University of Singapore and Nanchang University started to study the taxonomy and diversity of the freshwater crabs of Yunnan Province. Despite of the short period of the collection, 25 species of the freshwater crabs were collected, of which nine species were not be identified to any of described species (Jing En Chia, 2007, unpublished data). The present study investigated the identities of eight out of the above nine species by comparing type specimens, original descriptions and additional references of described allied species in detail. As a result, the eight species treated in this study turned out to be undescribed species of four genera, viz., Indochinamon Yeo & Ng, 2007, Pararanguna Dai & Chen, 1985, Parvuspotamon Dai & Bo, 1994, and Potamiscus Alcock, 1909. Indochinamon was established by Yeo & Ng (2007) for species distributed from southeast China (Yunnan and Guangxi), Vietnam, Laos, Thailand, Myanmar, northeastern India to Himalaya (Alcock, 1909; Pretzmann, 1966; Yeo & Ng, 1998; Dai, 1999; Brandis, 2000; Ng & Naiyanetr, 1993; Naiyanetr, 2001). They were previously assigned to “Potamon,” which was until then used as a “catch-all” genus for many Asian potamid species (Yeo & Ng, 2007). Recently four Vietnamese species have been added to the genus (Naruse, Quynh & Yeo, 2011; Do, Nguyen & Le, 2016). The present study describes additional four species that fit diagnostic characters of Indochinamon (see Yeo & Ng, 2007: 283) but differs from any known species. Poorly known Pararanguna and Parvuspotamon are both monotypic genera and endemic to Yunnan Province. One species each are described for these two genera, and the two genera are redefined. Potamiscus is distributed from China (Guangxi, Yunnan, Tibet), Myanmar to northeastern India (Wood-Mason, 1871; Rathbun, 1904; Alcock, 1909; Kemp, 1913; Türkay & Naiyanetr, 1987; Dai, 1999). Two new Potamiscus species are also described in this study.

The ninth species not treated in this study will be described as a new species of a new genus in a separate paper (Darren C. J. Yeo et al., unpublished data).

Methods

Specimens examined were collected by members of Chuxiong Medical College, Yunnan, and National University of Singapore for our study and are deposited in the Department of Parasitology, Medical College of Nanchang University, Nanchang (NCU MCP); the Zoological Reference Collection (ZRC), Lee Kong Chian Natural History Museum (previously Raffles Museum of Biodiversity Research), National University of Singapore; the Institute of Zoology, the Chinese Academy of Sciences, Beijing (CB); and the Ryukyu University Museum, Fujukan (RUMF), University of the Ryukyus, Okinawa. These collected specimens have been preserved in 75–80% ethanol. Measurements provided are of the carapace length (CL) by the carapace width (CW). Terminology used in descriptive accounts essentially follows Ng (1988) and Yeo & Ng (2007). The abbreviations G1 and G2 are used for the male first and second gonopods, respectively.

Morphology of some comparative species were previously not well-illustrated, which sometime makes difficult to conduct taxonomic work. A part of photographs taken by a team of the third author (XZ) were shown in this study. Data of those specimens were as follows: Indochinamon gengmaense (Dai, 1995), holotype, male (CB05192 YN6491119A), Mengding Town, Dima County Yunnan Province, coll. May 7, 1964. I. chinghungense (Dai et al., 1975), holotype, male (CB05166 YN 637507) (47.8 × 37.0 mm), Tuanshanzai Town Jinghong County, Yunnan Province, coll. December 9, 1963. I. boshanense (Dai & Chen, 1985), holotype, male (CB05160 HD8183031), Daojie Town, Baoshan County, Yunnan Province, coll. October 17, 1981. I. jianchuanense (Dai & Chen, 1985), holotype, male (CB05159 HD 8183030), Lingcheng Town, Jianchuan County, Yunnan Province, coll. September 28, 1981. I. menglaense (Dai & Cai, 1998), holotype, male (CB05168 YN-9496196A), (43.1 × 33.2 mm), Shangyong, Mengla County, Yunnan Province, coll., April 23–26, 1994. Pararanguna semilunata (Dai & Chen, 1985), paratype, male (CB05191 HD 8183034), Xiyi Town, Baoshan County, Yunnan Province, coll. October 13, 1981. Parvuspotamon yuxiense Dai & Bo, 1994, holotype, male (CB05138 YN 9091116A), Xingping, Yuxi County, Yunnan Province, coll. August, 1989. Potamiscus motuoensis Dai, 1990, holotype, male (CB05157 XZ6389084), Motuo County, Tibet, coll. July 29, 1983. Potamiscus yongshengensis Dai & Chen, 1985, holotype, male (CB05149 HD 8183035), Yongsheng County, Yunnan Province, coll. August 22, 1981. Other comparative material are as follows: I. boshanense (Dai & Chen, 1985), ZRC 1998.811, one male (50.1 × 36.8 mm), Boshan (Baoshan), Yunnan Province, China, coll. A.Y. Dai, October 20, 1981. Pararanguna semilunata Dai & Chen, 1985, one male (21.8 × 16.7 mm), one female (20.0 × 16.0 mm) (ZRC), Boshan (Baoshan), coll. October 13, 1981.

The electronic version of this article in portable document format will represent a published work according to the International Commission on Zoological Nomenclature (ICZN), and hence the new names contained in the electronic version are effectively published under that Code from the electronic edition alone. This published work and the nomenclatural acts it contains have been registered in ZooBank, the online registration system for the ICZN. The ZooBank LSIDs (Life Science Identifiers) can be resolved and the associated information viewed through any standard web browser by appending the LSID to the prefix http://zoobank.org/. The LSID for this publication is: urn:lsid:zoobank.org:pub:642C4FCB-905B-48DB-8E01-96C87249D809. The online version of this work is archived and available from the following digital repositories: PeerJ, PubMed Central and CLOCKSS.

Results

Systematics

Family Potamidae Ortmann, 1896

Subfamily Potamiscinae Ortmann, 1896 (sensu Yeo & Ng, 2003)

IndochinamonYeo & Ng, 2007

Potamon—Bott, 1967: 10 (part); Bott, 1970: 134 (part); Dai, 1999: 157 (not Potamon Savigny, 1816).

Potamon (Potamon)—Rathbun, 1904: 247 (part); Alcock, 1910: 19 (part).

Potamon (Himalayapotamon) Pretzmann, 1966: 4 (part).

Indochinamon Yeo & Ng, 2007: 282; Ng, Guinot & Davie, 2008: 163; De Grave et al., 2009: 40.

Diagnosis. Body size large (largest Chinese individual with CW 59.4 mm). Carapace relatively wide, low, with relatively flat dorsal surface, epigastric and postorbital cristae distinct, separated from each other by distinct groove, postorbital cristae not confluent with epibranchial tooth. Third maxilliped exopod with well-developed flagellum. Ambulatory legs relatively stout. Male sterno-pleonal cavity reaching imaginary line joining middle of bases of cheliped coxae. Male pleon narrowly triangular; G1 stout, terminal segment relatively short, stout, directed anterolaterally to laterally, lacking dorsal flap, with groove for G2 marginal in position.

Remarks. Indochinamon contains 38 species, including four new species described in the present study, from southeast China (Yunnan and Guangxi), Vietnam, Laos, Thailand, Myanmar, northeastern India to Himalaya. The type species of the genus is Potamon villosum Yeo & Ng, 1998, from northern Laos. Yeo & Ng (2007: 283) raised eight diagnostic characters for the genus: “(i) carapace low, with relatively flat dorsal surface; (ii) epigastric cristae separated from postorbital cristae by distinct groove; (iii) postorbital cristae not confluent with epibranchial tooth; (iv) third maxilliped exopod with well-developed flagellum; (v) ambulatory legs relatively stout; (vi) male pleon narrowly triangular; (vii) male sterno-pleonal cavity reaching imaginary line joining middle of bases of cheliped coxae; and (viii) G1 terminal segment relatively short and stout, with the groove for the G2 marginal in position, and lacking a dorsal flap.”

In the key to the genera of Chinese Potamidae, Dai (1999: 85, in Chinese) characterized Indochinamon (as Potamon) by the following characters: (1) the distance between mesial ends of thoracic sutures 4/5 is shorter than the one-third distance between sternal press-buttons of locking mechanism; (2) the G1 terminal segment tapers, without any protrusion; (3) the exopod of the third maxilliped has developed flagellum; and (4) G1 terminal segment is stout, conical. It is, however, difficult to apply the first character to Indochinamon species. For example, the proportion of the distance between mesial ends of thoracic suture 4/5 to the distance between sternal press-buttons of locking mechanism is smaller than one-third in I. ahkense sp. n. and I. tujiense, but that is larger than one-third in I. parpidum and I. lui. Inochinamon boshanense is also shown to have a larger value than one-third in this character (Dai, 1999: fig. 96 (3)). This is probably because the mesial ends of the sutures are not always clear and mesially continued gradually as depression, so it is difficult to standardize the measurement.

Indochinamon ahkense sp. n.

urn:lsid:zoobank.org:act:98393C16-DDA8-4A13-A9A3-C8778C17E8E6

Figs. 2–5.

Figure 2 Indochinamon ahkense sp. n. (Holotype, male, NCU MCP 2013.0003, 41.3 × 32.0 mm).

(A) Habitus, dorsal view; (B) cephalothorax, anterior view.

Figure 3 Indochinamon ahkense sp. n. (Holotype, male, NCU MCP 2013.0003, 41.3 × 32.0 mm).

(A) Cephalothorax, ventral view; (B) left chela, outer view.

Figure 4 Indochinamon ahkense sp. n. (Holotype, male, NCU MCP 2013.0003, 41.3 × 32.0 mm).

(A–C) Left G1; (A) ventral view; (B) enlarged view of distal portion, ventral view; (C) dorsal view; (D) left G2, dorsal view. Scales = 3 mm.

Figure 5 Indochinamon ahkense sp. n. (Paratype, female, ZRC 2013.0551, 43.1 × 33.2 mm).

Thoracic sternum with vulvae.

Material examined. Holotype, male (41.3 × 32.0 mm) (NCU MCP 2013.0003), Shaping Village, Ahke Town, Guangnan County, Yunnan Province, China, coll. Chen Zeng Long, February 1, 2004.

Paratypes: seven males (largest 33.8 × 26.3 mm), three females (largest 43.3 × 33.4 mm) (NCU MCP 2013.0004); four males (largest 38.4 × 29.7 mm), two females (larger 43.1 × 33.2 mm) (ZRC 2013.0551), two males (larger 35.6 × 27.4 mm), two females (larger 32.5 × 24.9 mm) (RUMF-ZC-2366), same data as holotype.

Diagnosis. Carapace (Fig. 2A) broader than long, CW 1.29–1.31 times (mean 1.30, n = 7) CL; dorsal surface (Fig. 2B) flat, regions demarcated, with short, indistinct setae on metabranchial region; cervical groove distinct, deep, reaching postorbital cristae; epigastric cristae rounded, not sharp, distinctly anterior to postorbital cristae, separated from postorbital cristae by distinct groove; postorbital cristae rounded, not reaching epibranchial tooth; regions behind epigastric and postorbital cristae rugose, branchial region weakly granulose; antennular fossae (Fig. 2B) rectangular in anterior view; external orbital angle broadly triangular, outer margin longer than inner margin, distinctly cristate, with shallow notch demarcating it from epibranchial tooth; epibranchial tooth distinct; anterolateral margin convex, serrated, distinctly cristate, confluent with posterolateral margin. Epistome (Fig. 2B) posterior margin median tooth well-developed, laterally sloping downward, only slightly arching. Third maxilliped (Figs. 2B and 3A) exopod with distinct flagellum exceeding two-thirds merus width. Ambulatory legs (Fig. 2A) with slender dactyli, carpi with well-defined median ridges; dactyli of last pair of ambulatory legs about same median length as propodi. Suture between sternites 3 and 4 (Fig. 3A) distinct, straight; sterno-pleonal cavity reaching imaginary line joining middle of bases of cheliped coxae. Male pleon (Fig. 3A) narrowly triangular; telson broadly triangular, proximal width about 1.2 times its median length; somite 6 trapezoidal, proximal width about 2.2 times median length. G1 (Figs. 4A–4C) terminal segment slender, subconical, tapered tip, slightly curved, bent obliquely outward (ca. <30°), with rather broad neck between terminal and subterminal segments; subterminal segment with distinct cleft on upper part of outer margin. Vulva (Fig. 5) oval, large, opening directed mesioventrally, located mesial end of suture 5/6, posterolateral end widely produced as short eave, opening covered with membrane.

Etymology. The species is named after the locality in which it was first found.

Remarks. I. ahkense sp. n. closely resembles I. gengmaense (Dai, 1995) in its general carapace morphology and in the slender, subconical, slightly curved G1 terminal segment, which has a tapered tip and is bent obliquely outward. However, I. ahkense sp. n. can be distinguished from I. gengmaense by the following characters: the external orbital angle is blunt, with a shallow notch demarcating it from the epibranchial tooth (vs. external orbital angle acute, with deep notch demarcating it from epibranchial tooth); and the border between the G1 terminal and subterminal segments is rather broad (vs. border narrow) (Figs. 2A, 3A, 4A–4C vs. Figs. 6A and 6F; Dai, 1995: fig. 5 (4, 5), pl. 1 fig. 5; Dai, 1999: fig. 98 (4, 5), pl. 12 fig. 1).

Figure 6 Dorsal views of habitus (A–E) and ventral views of G1s (F–J) of comparative Indochinamon species.

(A, F) I. gengmaense (Dai, 1995) (Holotype, CB05192 YN6491119A); (B, G) I. chinghungense (Dai et al., 1975) (Holotype, CB05166 YN 637507); (C, H) I. boshanense (Dai & Chen, 1985) (Holotype, CB05160 HD8183031); (D, I) I. jianchuanense (Dai & Chen, 1985) (Holotype, CB005159 HD 8183030); (E, J) I. menglaense (Dai & Cai, 1998) (Holotype, CB05168 YN-9496196A).

The new species is also similar to I. tannanti (Rathbun, 1904) and I. orleansi (Rathbun, 1904) in the shapes of the G1. Indochinamon ahkense sp. n. can be distinguished from I. tannanti by its flatter carapace (vs. slightly convex), having the epistome posterior margin divergent posteriorly (vs. almost straight), and the proportionally wider telson (vs. telson proportionally narrower) (Figs. 2A, 2B and 3A; vs. Yeo & Ng, 1998: fig. 1B and 1C). The new species can be differentiated from I. orleansi by its straight lateral margins of telson (vs. lateral margins concave), and sharp and distinct epibranchial tooth (vs. epibranchial tooth low) (Figs. 2A and 3A vs. Rathbun, 1904: fig. 20; Yeo & Ng, 1998: fig. 6B).

Distribution. Shaping Village, Ahke Town, Guangnan County, Yunnan Province, China.

Indochinamon parpidum sp. n.

urn:lsid:zoobank.org:act:FC1B9878-292C-4492-8A8D-44C7935A6904

Figs. 7–10.

Figure 7 Indochinamon parpidum sp. n. (Holotype, male, NCU MCP 2013.0015, 47.1 × 36.5 mm).

(A) Habitus, dorsal view; (B) cephalothorax, anterior view.

Figure 8 Indochinamon parpidum sp. n. (Holotype, male, NCU MCP 2013.0015, 47.1 × 36.5 mm).

(A) Cephalothorax, ventral view; (B) right chela, outer view.

Figure 9 Indochinamon parpidum sp. n. (Holotype, male, NCU MCP 2013.0015, 47.1 × 36.5 mm).

(A–D) Left G1; (A) ventral view; (B) enlarged view of distal portion, ventral view; (C) dorsal view; (D) enlarged view of distal portion, dorsal view; (E) left G2, ventral view. Scales = 3 mm.

Figure 10 Indochinamon parpidum sp. n. (Paratype, female, ZRC 2013.0558, 33.0 × 24.7 mm).

Thoracic sternum with vulvae.

Material examined. Holotype, male (47.1 × 36.5 mm) (NCU MCP 2013.0015), Niujie Town, Shiping County, Yunnan Province, China, coll. Li Hai Chun, February 23, 2004.

Paratypes: four males (largest 29.1 × 22.2 mm), 10 females (largest 45.3 × 33.0 mm) (NCU MCP 2013.0016), two males (larger 43.4 × 32.3 mm), two females (larger 35.0 × 26.2 mm) (ZRC 2013.0558), two males (larger 39.3 × 29.7 mm), two females (larger 33.5 × 25.6 mm) (RUMF-ZC-2371), same data as holotype.

Diagnosis. Carapace (Fig. 7A) broader than long, CW 1.29–1.37 times (mean 1.33, n = 7) CL; dorsal surface (Fig. 7B) flat; cervical groove distinct, relatively deep, reaching postorbital cristae; epigastric cristae strong, raised, rounded, distinctly anterior to postorbital cristae; postorbital cristae sharp, breaking out into granules just before reaching epibranchial tooth, slanting posterolaterally toward anterolateral margin, not confluent with epibranchial teeth; branchial region and regions behind epigastric and postorbital cristae rugose; frontal margin gently sinuous; antennular fossae rectangular (Fig. 7B) in anterior view; external orbital angle broadly triangular, with shallow, weak notch demarcating it from epibranchial tooth; epibranchial tooth present, very small, poorly developed; anterolateral margin convex, running inward posteriorly. Epistome (Fig. 7B) posterior margin with median tooth well-developed, acutely triangular; laterally sloping downward, almost straight. Third maxilliped (Figs. 7B and 8A) exopod with distinct flagellum about two-thirds merus width. Suture between sternites 3 and 4 (Fig. 8A) distinct, straight; lateral margins of sternite 4 straight; sterno-pleonal cavity reaching imaginary line joining middle of bases of cheliped coxae. Male pleon (Fig. 8A) narrowly triangular; telson broadly triangular, lateral margins distinctly concave; somite 6 trapezoidal, proximal width about 2.1 times its median length, lateral margins convex; lateral margins of somites 3 straight. G1 (Fig. 9) terminal segment short, subconical, with bulge of constant width that gradually tapers distally along inner margin, obliquely bent outward (ca. 45°), curved distally with narrowly tapered tip, groove for G2 marginal; subterminal segment gently sinuous, slender, with cleft on upper part of outer margin. Vulva (Fig. 10) inverted narrow triangular, opening directed mesioventrally, located posterior to mesial end of suture 5/6, lateral end widely produced as short eave, opening covered with membrane.

Etymology. The species name is an arbitrarycombination of par, Latin for “resemble,” with the species name I. hispidum (Wood-Mason, 1871), alluding to the similarity in the shape of their G1 terminal segments.

Remarks. Indochinamon parpidum sp. n. closely resembles I. hispidum (Wood-Mason, 1871), in general carapace morphology and the short, subconical G1 terminal segment, which is obliquely bent outward and curved distally, with a low, broad bulge on extensor margin. However, I. parpidum sp. n. can be distinguished from I. hispidum by the following characters: the bulge of the G1 terminal segment is gradually tapered distally along the extensor margin (vs. bulge narrow, with constant width along extensor margin); the G1 terminal segment is strongly and obliquely bent outward (ca. 45°), with a narrowly tapered tip (vs. G1 terminal segment obliquely bent outward to a lesser degree (ca. 30°), with a broadly tapered tip); the G1 subterminal segment has a cleft on the distal part of the outer margin (vs. G1 subterminal segment without cleft on upper part of outer margin), and the shape of the vulva is inverted narrow triangular (vs. vulva oval) (Figs. 9A–9C and 10 vs. Dai, 1999: fig. 94 (4, 5, 8)).

Dai & Chen (1985) described two subspecies of I. hispidum, i.e., I. h. jianchuanense (Dai & Chen, 1985) and I. h. boshanense (Dai & Chen, 1985). The former was distinguished from I. h. hispidum by its deeper grooves of the dorsal surface of the carapace, obtuse external orbital angle, a greater degree of the bent of the G1 terminal segment, and convex ventrolateral margin of the vulva, whereas the latter was differentiated from the type subspecies by a greater degree of the bent of the G1 distal segment. Dai & Chen (1985) indicated that the degree on the bent of the G1 terminal segment of I. h. boshanense is over 60°, but judging from the photograph of the G1 of the holotype (Fig. 6H) and the original description, its degrees against the mesial margin of the subterminal segment is ca. 80°. Although the two subspecies were erected to full species by Ng, Guinot & Davie (2008) without explanation, we consider the differences raised by Dai & Chen (1985) are significant to recognized them as full species.

Indochinamon parpidum sp. n. is morphologically similar to I. jianchuanense and I. boshanense. The new species can be distinguished from I. jianchuanense by the following characters: the extraorbital angle is sharp (vs. extraorbital angle obtuse), and the G1 terminal segment bends outward to a lesser degree (ca. 45°) with a narrowly tapered tip (vs. terminal segment with stronger bend, ca. 60°), and the G1 subterminal segment is more slender (vs. more stouter) (Figs. 7A, 9A vs. Figs. 6D and 6I; Dai & Chen, 1985: fig. 1 (4, 5), pl. 1 (1); Dai, 1999: fig. 95 (4, 5), pl. 11 (6)). The new species can also be distinguished from I. boshanense by the following characters: the G1 terminal segment bends outward to a lesser degree (ca. 45°) (vs. terminal segment with stronger bend, ca. 80°), and the G1 subterminal segment is slenderer (vs. stouter) (Figs. 7A and 9A vs. Figs. 6C and 6H; Dai & Chen, 1985: fig. 2 (4, 5), pl. 1 (2); Dai, 1999: fig. 96 (4, 5), pl. 11 (7)).

Indochinamon parpidum sp. n. is also morphologically similar to I. guttum (Yeo & Ng, 1998), but the former can be distinguished from the latter by the distal portion of the G1 subterminal segment being gradually bent laterally (vs. distal portion of G1 sutermnial segment abruptly bent laterally) (Figs. 9A–9C vs. Yeo & Ng, 1998: figs. 4K–4M).

Distribution. Niujie Town, Shiping County, Yunnan Province, China.

Indochinamon tujiense sp. n.

urn:lsid:zoobank.org:act:B1E8716A-894F-44E6-8D41-7226FAF9BE28

Figs. 11–14.

Figure 11 Indochinamon tujiense sp. n. (Holotype, male, NCU MCP 2013.0005, 45.3 × 33.2 mm).

(A) Habitus, dorsal view; (B) cephalothorax, anterior view.

Figure 12 Indochinamon tujiense sp. n. (Holotype, male, NCU MCP 2013.0005, 45.3 × 33.2 mm).

(A) Cephalothorax, ventral view; (B) right chela, outer view.

Figure 13 Indochinamon tujiense sp. n. (Holotype, male, NCU MCP 2013.0005, 45.3 × 33.2 mm).

(A–D) Left G1; (A) ventral view; (B) enlarged view of distal portion, ventral view; (C) dorsal view; (D) enlarged view of distal portion, dorsal view; (E) right G2, ventral view. Scales = 3 mm.

Figure 14 Indochinamon tujiense sp. n. (Paratype, NCU MCP 2013.0006, 35.6 × 26.7 mm).

Thoracic sternum with vulvae.

Material examined. Holotype, male (45.3 × 33.2 mm) (NCU MCP 2013.0005), Tujie Town, Nanhua County, Yunnan Province, China, coll. He Yong Gang, February 23, 2004.

Paratypes: three females (largest 35.6 × 26.7 mm) (NCU MCP 2013.0006), same data as holotype.

Diagnosis. Carapace (Fig. 11A) broader than long, CW 1.33–1.36 times (mean 1.35, n = 2) CL; dorsal surface (Fig. 11B) flat, short setae present on branchial and metabranchial regions, with regions distinctly demarcated; cervical groove distinct, relatively deep; epigastric cristae strong, raised, rounded, distinctly anterior to postorbital cristae, separated from postorbital cristae by shallow groove; postorbital cristae sharp, breaking out into granules just before reaching epibranchial tooth; branchial region and regions behind epigastric and postorbital cristae rugose; antennular fossae (Fig. 11B) rectangular in anterior view; external orbital angle broadly triangular, with distinct, shallow notch demarcating it from epibranchial tooth; epibranchial tooth distinct; anterolateral margin convex, granular, confluent with posterolateral margin; epistomal region narrow. Epistome (Fig. 11B) posterior margin with median tooth well-developed; laterally sloping downward, gently arched. Third maxilliped (Figs. 11B and 12A) exopod with distinct flagellum about half merus width. Ambulatory legs (Fig. 11A) with short, stout dactyli, carpi with sharply defined median ridges, meri rugose; dactyli of last pair of ambulatory legs with median length about 5.7 times its proximal width, about same median length as propodi. Suture between sternites 3 and 4 (Fig. 12A) distinct, shallow, medially concave; lateral margins of sternite 4 distinctly concave; sterno-pleonal cavity reaching imaginary line joining middle of bases of cheliped coxae. Male pleon (Fig. 12A) narrowly triangular; telson broadly triangular, with lateral margins gently concave; somite 6 trapezoidal, lateral margins straight; lateral margins of somite 3 distinctly concave. G1 (Figs. 13A–13D) terminal segment short, subconical, strongly bent obliquely outward (ca. 70°), about 0.24 times the length of subterminal segment, with tapering tip; extensor margin forming a bulge of constant width; outer margin sinuous, with a cleft along upper part of outer margin; subterminal segment with distinct cleft on upper part of outer margin, greatest width about 0.41 times its length. Vulva (Fig. 14) oval, large, opening directed mesioventrally, located mesial end of suture 5/6, lateral end widely produced as short eave, opening covered with membrane.

Etymology. The species is named after the locality in which it was first found.

Remarks. Indochinamon tujiense sp. n. closely resembles I. boshanense (Dai & Chen, 1985), in its general carapace morphology and in the short, subconical G1 terminal segment being strongly bent obliquely outward, with a slight hump along the outer margin at the proximal end and a sinuous outer margin. However, I. tujiense sp. n. can be distinguished from I. boshanense by the following characters: the epigastric cristae is strong (vs. epigastric cristae relatively weak); the lateral margins of male telson are gently concave (vs. lateral margins of male telson distinctly and widely concave); the G1 terminal segment has a broad bulge with constant width along the extensor margin (vs. G1 terminal segment with a relatively narrow bulge along the extensor margin), and the shape of the vulva is oval (vs. vulva narrow transversely) (Figs. 11A, 12A, 13 and 14 vs. Fig. 4C; Dai & Chen, 1985: fig. 2 (4, 5, 7, 9), pl. 1 fig. 2; Dai, 1999: fig. 96 (2, 4, 5, 8), pl. 11 fig. 7).

The new species is also morphologically similar to I. jianchuanense (Dai & Chen, 1985), but the former can be differentiated from the latter by the greater degree of the bent of the G1 terminal segment (ca. 70°) (vs. bent to a lesser degree, ca. 60°); and the relatively larger bulge on the extensor margin of G1 terminal segment (vs. bulge narrower); and the oval vulval opening (vs. the transversely narrow vulval opening) (Figs. 13 and 14 vs. Fig. 4I; Dai & Chen, 1985: fig. 1 (4, 5, 9); Dai, 1999: fig. 95 (4, 5, 8)).

The new species can be distinguished from I. menglaense (Dai & Cai, 1998) by the anterolateral margin of the carapace being lined with rounded granules (vs. anterolateral margin lined with spine-like granules); the slightly concave lateral margins of male telson (vs. distinctly concave); the constantly wide bulge on the extensor margin of G1 terminal segment (vs. bulge more produced, rounded); and the oval vulval opening (vs. transversely wide vulval opening) (Figs. 11A, 12A, 13 and 14 vs. Figs. 6E, 6J; Dai & Cai, 1998: figs. 12, 14, 15 and 18).

The new species differs from I. boshanense in the following characters: the mesial margin of the G1 subterminal segment is more produced subdistally in dorsal view (vs. mesial margin straight, oblique, not produced subdistally); the G1 terminal segment bends to a lesser degree (ca. 70°) (vs. terminal segment with stronger bend, ca. 80°); the bulge on the extensor margin of the G1 terminal segment is larger (vs. smaller); and the shape of the vulva is oval (vs. vulva transversely narrow) (Figs. 13 and 14 vs. Fig. 6H; Dai & Chen, 1985: fig. 2 (4, 5, 9); Dai, 1999: fig. 96 (4, 5, 8)).

The new species also differs from I. dangi Naruse, Quynh & Yeo, 2011, in the following characters: the infraorbital margin is straight mesially and curves upward laterally (vs. infraorbital margin almost entirely straight and oblique); the G1 terminal segment bends to a lesser degree (ca. 70°) (vs. terminal segment with stronger bend, ca. 90°); and the bulge on the extensor margin of the G1 terminal segment is constant in width (vs. bulge absent) (Figs. 11B and 13 vs. Naruse, Quynh & Yeo, 2011: figs. 9d, 9e and 11a).

Distribution. Tuzie Town, Nanhua County, Yunnan Province, China.

Indochinamon lui sp. n.

urn:lsid:zoobank.org:act:3FAF5B0F-9963-4F00-9DBD-5FDE982DD04E

Figs. 15–18.

Figure 15 Indochinamon lui sp. n. (Holotype, male, NCU MCP 2013.0010, 43.0 × 32.9 mm).

(A) Habitus, dorsal view; (B) cephalothorax, anterior view.

Figure 16 Indochinamon lui sp. n. (Holotype, male, NCU MCP 2013.0010, 43.0 × 32.9 mm).

(A) Cephalothorax, ventral view; (B) right chela, outer view.

Figure 17 Indochinamon lui sp. n. (Holotype, male, NCU MCP 2013.0010, 43.0 × 32.9 mm).

(A–D) Left G1; (A) ventral view; (B) enlarged view of distal portion, ventral view; (C) dorsal view; (D) enlarged view of distal portion, dorsal view; (E) right G2, ventral view. Scales = 3 mm.

Figure 18 Indochinamon lui sp. n. Paratype, RUMF-ZC-2369, 34.8 × 26.2 mm).

Thoracic sternum with vulvae.

Material examined. Holotype, male (43.0 × 32.9 mm) (NCU MCP 2013.0010), Mang Huai Town, Yun County, Yunnan Province, China, coll. Lu Yong Feng, February 24, 2004.

Paratypes: two males (larger 30.8 × 23.3 mm), one female (36.9 × 27.5 mm) (NCU MCP 2013.0011), two males (larger 33.1 × 25.0 mm), one female (31.4 × 23.6 mm) (ZRC 2013.0555), one male (25.5 × 19.6 mm), one female (34.8 × 26.2 mm) (RUMF-ZC-2369), same data as holotype.

Others: one male (40.3 × 30.2 mm), six females (largest 36.2 × 28.1 mm), five juveniles (NCU MCP 2013.0012), two males (larger 32.2 × 24.9 mm) (ZRC 2013.0556), Dashan Village, Xueshan Town, Fengqing County, Yunnan Province, China, coll. Yang Zheng Bing, February 1, 2004; nine males (largest male 33.3 × 24.4 mm), 13 females (largest 41.7 × 31.2 mm) (NCU MCP 2013.0013), Xinfu Town, Yun County, Yunnan Province, China, coll. Shen Tian Juan, February 26, 2004; seven males (largest 39.3 × 29.9 mm), nine females (largest 41.0 × 30.2 mm), three juveniles (NCU MCP 2013.0014), two males (larger 37.5 × 28.8 mm), two females (larger 38.6 × 28.6 mm) (ZRC 2013.0557), three males (largest 34.6 × 25.7 mm), two females (larger 38.5 × 29.0 mm) (RUMF-ZC-2370), Mongku Town, Shuangjiang County, Yunnan Province, China, coll. Li Quan Cheng, February 26, 2004.

Diagnosis. Carapace (Fig. 15A) broader than long, CW 1.29–1.36 times (mean 1.33, n = 18) CL; dorsal surface (Fig. 15B) flat, sparse and short setae on branchial and metabranchial regions, with regions distinctly demarcated; cervical groove distinct, relatively deep, reaching postorbital cristae; epigastric cristae rounded, distinctly anterior to postorbital cristae, separated from postorbital cristae by shallow groove; postorbital cristae sharp, breaking out into few, weak granules just before reaching epibranchial tooth; branchial region and regions behind epigastric and postorbital cristae slightly rugose; antennular fossae (Fig. 15B) rectangular in anterior view; external orbital angle broadly triangular, with outer margin longer than inner margin, with distinct, shallow notch demarcating it from epibranchial tooth; epibranchial tooth distinct; anterolateral margin convex, confluent with posterolateral margin. Epistome (Fig. 15B) posterior margin with median tooth well-developed, obtusely triangular; laterally sloping downward. Third maxilliped (Figs. 15B and 16A) exopod with distinct flagellum about two-thirds merus width. Ambulatory legs (Fig. 15A) with short, stout dactyli, carpi with sharply defined ridges, meri rugose; dactyli of last pair of ambulatory legs with median length about 6.5 times its proximal width, about 1.1 times as long as its propodus. Suture between sternites 3 and 4 (Fig. 16A) distinct; lateral margins of sternite 4 slightly concave anteriorly; sterno-pleonal cavity just exceeding imaginary line joining middle of bases of cheliped coxae. Male pleon (Fig. 16A) narrowly triangular; telson broadly triangular, with lateral margins gently concave; somite 6 trapezoidal, lateral margins straight; lateral margins of somite 3 gently concave. G1 (Fig. 17) terminal segment short, subconical, with a bulge of increasing distal width along the extensor margin, bent obliquely outward (ca. 70°), with straight tapering tip, without dorsal flap. Vulva (Fig. 18) oval, large, opening directed mesioventrally, located mesial end of suture 5/6, lateral end widely produced as short eave, opening covered with membrane.

Etymology. The species is named after its collector, Lu Yong Feng.

Remarks. Indochinamon lui sp. n. closely resembles I. boshanense (Dai & Chen, 1985), in general carapace morphology and in the short, subconical G1 terminal segment, which is obliquely bent outward, with a tapering tip and without a dorsal flap. However, I. lui sp. n. can be distinguished from I. boshanense by the following characters: the epibranchial tooth is distinct (vs. epibranchial tooth poorly developed); the G1 terminal segment has the straight, tapering tip and a bulge of increasing distal width along the extensor margin (vs. G1 terminal segment with slightly curved, tapering tip and narrow bulge of constant width along the extensor margin); and the shape of the vulva is oval (vs. vulva transversely narrow) (Figs. 15A, 17 and 18 vs. Dai & Chen, 1985: fig. 2 (4, 5, 9), pl. 1 fig. 2; Dai, 1999: fig. 96 (4, 5, 8), pl. 11 fig. 7).

The new species is morphologically also similar to I. jianchuanense (Dai & Chen, 1985), but the former can be distinguished from the latter by the following characters: the G1 terminal segment is shorter and stouter (vs. G1 terminal segment longer and slender); the bulge on the extensor margin of the G1 terminal segment is more produced distally (vs. bulge more gradually narrowed distally); and the shape of the vulva is oval (vs. vulva transversely narrow) (Figs. 15A, 17 and 18 vs. Fig. 6I; Dai & Chen, 1985: fig. 1 (4, 5, 9); Dai, 1999: fig. 95 (4, 5, 8)).

The new species can be differentiated from I. dangi Naruse, Quynh & Yeo, 2011, by the lesser degree of the bend of the G1 terminal segment (ca. 70°) (vs. terminal segment with stronger bend, ca. 90°); and the bulge on the extensor margin of G1 terminal segment with constant width (vs. bulge absent) (Fig. 17 vs. Naruse, Quynh & Yeo, 2011: fig. 9d and 9e).

Distribution. Yun, Fengqing and Shuangjiang Counties, Yunnan Province, China. The collection site is located in the Lincang and Lancang Rivers Natural Reserve.

PararangunaDai & Chen, 1985

Ranguna (Pararanguna) Dai & Chen, 1985: 64.

Pararanguna—Dai, 1999: 370; Ng, Guinot & Davie, 2008: 165; De Grave et al., 2009: 40.

Diagnosis. Body size relatively small (largest individual with CW 22.0 mm). Carapace relatively high, with gently convex dorsal surface, epigastric and postorbital cristae very low, not continuous, postorbital cristae not confluent with epibranchial tooth. Third maxilliped exopod without flagellum. Ambulatory legs relatively slender. Male sterno-pleonal cavity reaching imaginary line joining middle of bases of cheliped coxae. Male pleon broadly triangular. G1 relatively slender, straight to gently sinuous; terminal segment stout, straight, length about 0.6–0.8 time that of subtemirnal segment, tip truncate, ventral layer produced outward to form large semicircular dorsal flap, groove for G2 marginal.

Remarks. Dai & Chen (1985) used the name Ranguna (Pararanguna) semilunatum consistently in the paper. The inclination is to assume the gender of the subgenus is neuter as the species named is neuter. The Latin “lunatum” means “crescent-shaped” and the gender is neuter (masculine “lunatus,” feminine “lunata”). The genus Ranguna Bott, 1966, on the other hand, is clearly feminine even though Bott (1966) did not specify the gender when he recognized it as a subgenus of Potamiscus Alcock, 1909. We, however, cannot extrapolate from Dai & Chen’s (1985) paper that they wanted the gender of Pararanguna to be neuter; regardless of the fact that it should be also feminine since it was based on Ranguna. The reason is that they only established Pararanguna as a subgenus of Ranguna; and the Code (International Commission on Zoological Nomenclature (ICZN), 1999) mandates that the gender of the species must match that of the genus—in this case, Ranguna, which already is feminine. Dai & Chen’s (1985) used of the neuter “semilunatum” must thus be regarded as a lapsus. The name “semilunatum” also cannot be regarded as a noun, it was neither stated nor etymologically correct. As such, it must be regarded as an adjective. In the present classification were Pararanguna is treated as a full genus (Ng, Guinot & Davie, 2008), it is thus necessary to treat this name as derived from the feminine Ranguna, i.e., Pararanguna is also feminine. As such, the gender of the constituent species is also feminine, i.e., Pararanguna semilunata.

When Pararanguna was established, Dai & Chen (1985), compared it with Ranguna and listed two diagnostic characters, i.e., (1) more prominent post-frontal lobe and post-orbital cristae, and (2) the first pleopod being tapered and directed outward, but they did not specify which genus has these character states. Specimens of Pararanguna semilunata examined in this study, including paratypes, have very low post-frontal crista and post-orbital crista (Fig. 19A), and subconical G1 terminal segment, with a bulge of increasing distal width along the extensor margin (Fig. 19E). This indicates that the two diagnostic characters listed by Dai & Chen (1985) were of Ranguna sensu Bott (1968, 1970).

Figure 19 Dorsal views of habitus (A–D) and ventral views of G1s (E–H) of comparative species.

(A, E) Pararanguna semilunata (Dai & Chen, 1984) (Holotype, CB05191 HD8183034); (B, F) Parvuspotamon yuxiense Dai & Bo, 1994 (Holotype, CB05138 YN 9091116A); (C, G) Potamiscus motuoensis Dai, 1990 (Holotype, CB05157 XZ6389084); (D, H) Potamiscus yongshengensis Dai & Chen, 1985 (Holotype, CB05149 HD 8183035). Photo credit: Chunchao Zhu and Jiexin Zou.

The genus Ranguna had taxonomic problems. Ranguna was established by Bott (1966) with Potamon (Potamon) rangoonensis Rathbun, 1904, as type species, although he did not examine the type specimen (see Türkay & Naiyanetr, 1989; Ng, 1990; Holthuis, 1990). Bott (1970) included 16 species and two subspecies in Ranguna. Türkay & Naiyanetr (1987) redescribed the holotype of Potamon rangoonensis, the type species of Ranguna, and found that it is not what they thought of “Ranguna” sensu Bott (1968, 1970) that was characterized by the presence of a dorsal fold on the terminal segment of the G1 and the G1 subterminal segment being not-noticeably narrowed distally. Potamon rangoonensis is now placed in Potamiscus Alcock, 1909, therefore Ranguna is a junior synonym of Potamiscus (see International Commission on Zoological Nomenclature (ICZN), 1991), whereas other ex-Ranguna species have been transferred to 13 genera, i.e., Badistemon Yeo & Ng, 2007, Dromothelphusa Naiyanetr, 1992; Eosamon Yeo & Ng, 2007, Hainanpotamon Dai, 1995, Iomon Yeo & Ng, 2007, Pilosamon Ng, 1996, Planumon, Yeo & Ng, 2007, Pupamon, Yeo & Ng, 2007, Stelomon Yeo & Naiyanetr, 2000, Stoliczia Bott, 1966, Thaipotamon Ng & Naiyanetr, 1993, Vietopotamon Dang & Ho, 2002, and Villopotamon Dang & Ho, 2003 (Ng, Guinot & Davie, 2008). The G1 of Pararanguna clearly differs from those of all “Ranguna (Ranguna)” species recognized in Bott (1970) in its truncate tip (vs. tip tapering) (Bott, 1970; Bott & Türkay, 1977; Ng, 1988, 1991; Ng & Naiyanetr, 1993; Brandis, 2000, 2002; Yeo & Naruse, 2007; Yeo & Ng, 2007; Dang & Ho, 2008; Yeo, 2010).

In Dai’s (1999) key to genera of Chinese Potamidae (p. 85, in Chinese), Pararanguna was grouped with Aparapotamon Dai & Chen, 1985, Tenuipotamon Dai, 1999, and Parvuspotamon Dai & Bo, 1994, by the following characters: (1) the distance between mesial ends of thoracic suture 4/5 is about equal to the one-third distance between sternal press-buttons of locking mechanism, (2) the terminal segment of G1 is longer than half the length of G1 subterminal segment; (3) the exopod of the third maxilliped lacks flagellum. The above first character in Pararanguna semilunatum is 35.6% (calculated from Dai, 1999: fig. 200 (3)), but this is difficult to determin whether it is “about equal to the one-third.” This character is also discussed for Indochinamon (see remarks of Indochinamon). The differences between above four genera are very distinct; the terminal segment of Pararanguna has a roundly truncate tip and well-developed semicircular dorsal flap (vs. terminal segment cylindrically stick-shaped without dorsal flap in Aparapotamon, terminal segment gently incurved with distal dorsal lobe larger than ventral lobe in Tenuipotamon, terminal segment tapered and distally curves inward in Parvuspotamon).

Pararanguna hemicyclia sp. n.

urn:lsid:zoobank.org:act:8D4F20E5-B08C-41C1-91AC-DADE13A58725

Figs. 20–23.

Figure 20 Pararanguna hemicyclia sp. n. (Holotype, male, NCU MCP 2013.0017, 14.3 × 12.5 mm).

(A) Habitus, dorsal view; (B) cephalothorax, anterior view. Photo credit: Tohru Naruse.

Figure 21 Pararanguna hemicyclia sp. n. (Holotype, male, NCU MCP 2013.0017, 14.3 × 12.5 mm).

Cephalothorax, ventral view. Photo credit: Tohru Naruse.

Figure 22 Pararanguna hemicyclia sp. n. (Holotype, male, NCU MCP 2013.0017, 14.3 × 12.5 mm).

(A–C) Right G1; (A) ventral view; (B) dorsal view; (C) lateral view; (D) right G2, dorsal view. Scale = 1 mm. Drawing credit: Jing En Chia.

Figure 23 Pararanguna hemicyclia sp. n. (Paratype, female, NCU MCP 2013.0018, 15.9 × 13.4 mm).

Thoracic sternum with vulvae. Photo credit: Tohru Naruse.

Material examined. Holotype, male (14.3 × 12.5 mm) (NCU MCP 2013.0017), Dashan Village, Xueshan Town, Fengqing County, Yunnan Province, China, coll. Yang Zheng Bing, February 1, 2004.

Paratypes: 11 males (largest 10.6 × 9.5 mm), 10 females (largest 16.2 × 13.6 mm) (NCU MCP 2013.0018), three males (largest 14.0 × 11.9 mm), two females (larger 15.1 × 12.8 mm) (ZRC 2013.0559), two males (larger 16.3 × 13.5 mm), two females (larger 15.1 × 12.4 mm) (RUMF-ZC-2372), same data as holotype.

Others: 20 males (largest 12.0 × 10.6 mm), 24 females (largest 15.1 × 12.9 mm) (NCU MCP 2013.0019), two males (larger 12.3 × 10.7 mm), one female (12.9 × 10.9 mm) (ZRC 2013.0560), one male (12.3 × 10.9 mm), one female (12.6 × 10.5 mm) (RUMF-ZC-2373), Fengqing County, Yunnan Province, China, coll. Yang Zheng Bing, February, 2004.

Diagnosis. Carapace (Fig. 20A) broader than long, CW 1.12–1.22 times (mean 1.17, n = 13) CL; dorsal surface (Fig. 20B) gently convex, with regions distinctly demarcated; cervical groove present, shallow, weakly developed; epigastric cristae rounded, flat, distinctly anterior to postorbital cristae; postorbital cristae weakly developed, gently sloping downward anteriorly, confluent with branchial and regions behind it; branchial region and regions behind epigastric and postorbital cristae rugose; antennular fossae (Fig. 20B) subtriangular in anterior view; external orbital angle acutely triangular, outer margin convex and longer than inner margin, with obvious cleft demarcating it from epibranchial tooth; epibranchial tooth distinct, small; anterolateral margin gently convex, confluent with posterolateral margin. Epistome (Fig. 20B) posterior margin with well-developed, triangular median tooth that has a pointed median tip. Third maxilliped (Figs. 20B and 21) exopod without flagellum. Carpi of chelipeds with well-developed, acute spines on inner margin. Ambulatory legs (Fig. 20A) hairy, with very long, slender dactyli. Suture between sternites 3 and 4 (Fig. 21) indistinct; sterno-pleonal cavity reaching imaginary line joining middle of bases of cheliped coxae. Male pleon (Fig. 21) broadly triangular; telson broadly triangular, lateral margins gently convex; somite 6 trapezoidal, lateral margins convex, proximal width about 2.8–3 times its median length. G1 (Figs. 22A–AC) relatively slender, gently sinuous; terminal segment stout, straight, length about 0.6 time that of subtemirnal segment, tip truncate, ventral layer produced outward to form semicircular dorsal flap, dorsal flap well-developed, high, broad, but proximally not reaching to proximal end of terminal segment, groove for G2 marginal. G2 (Fig. 22D) terminal segment cylindrical, truncate, less than half the length of basal segment. Vulva (Fig. 23) circular, large, opening directed ventrally, located mesial end of suture 5/6, opening covered with membrane.

Etymology. The species name is derived from hemicyclium, Latin for semicircle, alluding to the dorsal flap of the G1 terminal segment.

Remarks. Pararanguna hemicyclia sp. n. closely resembles the only congener, Pararanguna semilunata (Dai & Chen, 1985), in its general carapace morphology and in the stout, straight G1 terminal segment which is rounded with a slight subdistal constriction, possessing a truncate tip and a well-developed, high, broad dorsal flap. However, P. hemicyclia can be distinguished from Pararanguna semilunata by the following suite of diagnostic characters: the anterolateral margin of the carapace is less convex laterally (vs. anterolateral margin strongly convex laterally); the dorsal surface is gently convex with distinctly demarcated regions (vs. dorsal surface convex and inflated with weakly demarcated regions); the antennular fossae appear subtriangular in anterior view (vs. antennular fossae rectangular in anterior view); the external orbital angle is acutely triangular with an obvious cleft demarcating it from the epibranchial tooth (vs. external orbital angle obtusely triangular with a slight cleft demarcating it from epibranchial tooth); the median tooth of the epistome posterior margin has an acute tip (vs. median tooth with an obtuse tip); the carpi of the chelipeds have well-developed, acute spines on the inner margin (vs. chelipeds carpi with low, obtuse spines on inner margin); the male pleon is broadly triangular with telson having gently convex lateral margins (vs. male pleon relatively narrowly triangular with telson having almost straight lateral margins); the dorsal flap of the G1 terminal segment has the median part being higher and the proximal end of the dorsal flap does not reach the proximal end of the distal segment (vs. dorsal flap with a lower apex in median part and proximal end of dorsal flap reaching proximal end of distal segment). (Figs. 20, 21 and 22A–22C vs. Dai & Chen, 1985: fig. 16 (4, 5, 7), pl. 1 fig. 5; Dai, 1999: fig. 200, pl. 25 fig. 1).

Distribution. Dashan Village, Xueshan Town, Fengqing County, Yunnan Province, China. The collection site is located in the Lincang and Lancang Rivers Natural Reserve.

ParvuspotamonDai & Bo, 1994

Parvuspotamon Dai & Bo, 1994: 24; Dai, 1999: 406; Ng, Guinot & Davie, 2008: 165; De Grave et al., 2009: 40.

Diagnosis. Body size relatively small (largest individual with CW 25.7 mm). Carapace relatively high, with gently convex dorsal surface, epigastric cristae weak, postorbital cristae faint, not continuous, postorbital cristae not confluent with epibranchial tooth. Third maxilliped exopod without flagellum. Ambulatory legs relatively slender. Male sterno-pleonal cavity reaching imaginary line joining proximal portions of bases of cheliped coxae. Male pleon narrowly triangular G1 slender, sinuous; terminal segment slender, length, 0.4–0.6 time that of subterminal segment, distally digitiform, with truncate, narrowly rounded tip ending with dorsal apeature (visible from ventral view), with groove for G2 marginal.

Remarks. Dai & Bo (1994) compared Parvuspotamon with Yarepotamon Dai & Türkay, 1997, which has been known from Guangdon and Guanxi Provinces. They distinguished the two genera by the following characters: the carapace is more glabrous (vs. finely rugose in Yarepotamon), the postorbital criste are blunt (vs. sharp in Yarepotamon), the epibranchial tooth is blunt (vs. prominent in Yarepotamon), the exopod of the third maxilliped lacks a flagellum (vs. absent or with vestigial flagellum in Yarepotamon), and a groove for G2 on G1 placed laterally (vs. medially in Yarepotamon). Furthermore, they recognized more morphological differences between Parvuspotamon yuxiense and Yarepotamon gracilipa in the length to width ratio of the telson (1.1 vs. 1.4), and the G1 subterminal segment being 1.8 times as long as terminal segment (vs. 2.1 in Y. gracilipa), and the G2 subterminal segment is about 1.4 times as long as terminal segment (vs. two times in Y. gracilipa).

Parvuspotamon is morphologically more similar to Pararanguna, Aparapotamon, and Tenuipotamon, which are exclusively or partially distributed in Yunnan Provice, in their low to very low postorbital and postfrontal cristae and the absence of the flagellum from the third maxilliped exopod. These genera can, however, be easily distinguished from each other by the condition of the G1 terminal segment (see Remarks of Pararanguna).

According to the key to genera of Chinese Potamidae (Dai, 1999: 85, in Chinese), the abovementioned four Yunnan genera are supposed to have relatively long G1 distal segment (more than half the length of that of subteminal segment). This key should be amended to accommodate Parvuspotamon dixuense sp. n., G1 terminal segment of which is slightly shorter than half the length of the G1 subterminal segment (ca. 0.4). This change is also necessary for Tenuipotamon, as Tenuipotamon huaningense has relatively short G1 terminal segment (Dai & Bo, 1994: fig. 2 (4); Dai, 1999: fig. 206 (4)).

Parvuspotamon dixuense sp. n.

urn:lsid:zoobank.org:act:1D4DBFB9-B5BB-49CB-90EA-4C862CD75859

Figs. 24–27.

Figure 24 Parvuspotamon dixuense sp. n. (Holotype, male, NCU MCP 2013.0020, 25.7 × 20.0 mm).

(A) Habitus, dorsal view; (B) cephalothorax, anterior view.

Figure 25 Parvuspotamon dixuense sp. n. (Holotype, male, NCU MCP 2013.0020, 25.7 × 20.0 mm).

(A) Cephalothorax, ventral view; (B) right chela, outer view.

Figure 26 Parvuspotamon dixuense sp. n. (Holotype, male, NCU MCP 2013.0020, 25.7 × 20.0 mm).

(A–D) Right G1; (A) ventral view; (B) enlarged view of distal portion, ventral view; (C) dorsal view; (D) enlarged view of distal portion, dorsal view; (E) right G2, dorsal view. Scales = 1 mm.

Figure 27 Parvuspotamon dixuense sp. n. (Paratype, female, NCU MCP 2013.0021, 23.1 ×17.8 mm).

Thoracic sternum with vulvae.

Material examined. Holotype, male (25.7 × 20.0 mm) (NCU MCP 2013.0020), Huaguo Village, Dixu Town, Guangnan County, Yunnan Province, China, coll. Nong Guang Lin, February 25, 2004.

Paratypes: eight males (largest 23.3 × 17.4 mm), 10 females (largest 23.3 × 17.8 mm) (NCU MCP 2013.0021), five males (largest 24.7 × 18.9 mm), one female (21.3 × 16.2 mm) (ZRC 2013.0561), five males (largest 23.6 × 18.1 mm), one female (19.3 × 14.6 mm) (RUMF-ZC-2374), same data as holotype.

Others: two males (larger 16.7 × 11.9 mm), two females (larger 17.2 ×12.5 mm) (NCU MCP 2013.0022), Zhetu Village, Guangnan County, Yunnan Province, China, coll. unknown, November 2, 2002; one male (16.2 × 12.5 mm), one female (15.8 × 12.1 mm) (NCU MCP 2013.0023), Xiyangjiang, base of Jiulongshan, Zhetu Village, Guangnan County, Yunnan Province, China, coll. unknown, November 2, 2002.

Diagnosis. Carapace (Fig. 24A) broader than long, CW 1.19–1.40 times (mean 1.32, n = 11) CL; dorsal surface (Fig. 24B) gently convex, with regions distinctly demarcated; cervical groove indistinct, shallow; epigastric cristae rounded, weakly developed, separated from postorbital cristae by shallow groove; postorbital cristae rounded, fairly straight, reaching epibranchial tooth; branchial region and regions behind postorbital cristae weakly rugose; antennular fossae (Fig. 24B) narrowly rectangular in anterior view; external orbital angle triangular, low, outer margin gently convex, with distinct, shallow notch demarcating it from epibranchial tooth; epibranchial tooth weakly developed, small, granular, appears part of anterolateral margin; anterolateral margin convex, granular, confluent with posterolateral margin. Epistome (Fig. 24B) posterior margin with median tooth broadly triangular. Third maxilliped (Figs. 24B and 25A) exopod without flagellum. Ambulatory legs (Fig. 24A) with long, slender dactyli, carpi of first to third ambulatory legs with weak median ridges, meri slightly rugose. Suture between sternites 3 and 4 (Fig. 25A) weakly developed; sterno-pleonal cavity reaching imaginary line joining proximal part of bases of cheliped coxae. Male pleon (Fig. 25A) narrowly triangular; telson broadly triangular, proximal width about 1.3 times its median length, with lateral margins convex. G1 (Figs. 26A–26D) terminal segment long, slender, subconical, outer margin convex, length ca. 0.4 times that of subterminal segment, gently bent but not curving obliquely outward, inner margin straight, with truncate, narrowly rounded tip ending with dorsal hole (visible from ventral view), with groove for G2 marginal. G2 (Fig. 26E) subconical, terminal segment less than half the length of basal segment. Vulva (Fig. 27) oval, large, opening directed mesioventrally, located posterior to mesial end of suture 5/6, lateral end widely produced as short eave, opening covered with membrane.

Etymology. The species is named after the locality in which it was found.

Remarks. Parvuspotamon dixuense sp. n. closely resembles the only congener, P. yuxiense Dai & Bo, 1994, in its general carapace morphology and in the long, slender, subconical G1 terminal segment with curved outer margin, truncate tip and slender subterminal segment. However, P. dixuense can be distinguished from P. yuxiense by the following characters: the G1 terminal segment is gently bent outward, with a straight inner margin (vs. G1 terminal segment curved inward, with a concave inner margin); the groove for G2 runs along the ventral side of the G1 terminal segment (vs. groove for G2 being marginal along the G1 terminal segment); a shelf is present at the lateral margin between the G1 terminal and subterminal segments (vs. shelf absent); and the G2 terminal segment is less than half the length of the basal segment (G2 terminal segment more than half the length of the basal segment) (Fig. 26 vs. Fig. 19F; Dai & Bo, 1994: fig. 3 (4–6); Dai, 1999: fig. 216 (4–6)).

Distribution. Xiyangjiang, Zhetu and Huaguo Village, Dixu Town, Guangnan County, Yunnan Province, China.

PotamiscusAlcock, 1909

Potamiscus Alcock, 1909: 246; Bott, 1966: 479; Bott, 1970: 158; Dai, 1999: 186; Ng, Guinot & Davie, 2008: 165; De Grave et al., 2009: 40.

Potamon (Potamiscus)—Alcock, 1910: 56.

Potamon (Ranguna) Bott, 1966: 481.

Diagnosis. Body size midium (largest Chinese individual with CW 34.4 mm). Carapace relatively narrow, high, with gently convex dorsal surface, epigastric cristae distinct, postorbital cristae low but clear, not continuous, postorbital cristae not confluent with epibranchial tooth. Third maxilliped exopod without or with vestigial flagellum. Ambulatory legs relatively stout. Male sterno-pleonal cavity reaching imaginary line joining middle to anterior portions of bases of cheliped coxae. Male pleon narrowly triangular; G1 stout, subconical, terminal segment relatively short, stout, distally tapering to roundly tapering, lacking dorsal flap, with groove for G2 marginal in position.

Remarks. Potamiscus contains 16 species, including two new species described in the present study, from China (Guangxi, Yunnan, Tibet), Myanmar, and northeastern India. The type species of the genus is Potamon (Potamiscus) annandali Alcock, 1909, from Assam, northwestern India. Potamiscus is diagnosed by the following characters: (1) carapace with moderate height, dorsal surface rugose; (2) third maxilliped without flagellum or with vestigial flagellum; (3) G1 subconical, with distally tapering tip; (4) G1 terminal segment without dorsal flap; and (5) groove for G2 on G1 terminal segment marginal (Wood-Mason, 1871; Rathbun, 1904; Alcock, 1909; Kemp, 1913; Türkay & Naiyanetr, 1987; Dai, 1999). The two species described in this paper share these characters.

In the key to genera of Chinese Potamidae (Dai, 1999: 85, in Chinese), one of the characters that characterizes Potamiscus is: the distance between mesial ends of thoracic suture 4/5 is shorter than the one-third distance between sternal press-buttons of locking mechanism. However, the mesial end of the suture 4/5 is placed near the base of lateral slope of the sterno-pleonal cavity, which is nearly vertical, and the mesial end is not always clear and mesially continued gradually as depression, it is difficult to standardize the measurements.

Potamiscus fumariatus sp. n.

urn:lsid:zoobank.org:act:0A9F4D94-7238-4780-8171-9D0D6DF36881

Figs. 28–30.

Figure 28 Potamiscus fumariatus sp. n. (Holotype, male, NCU MCP 2013.0024, 24.1 × 20.1 mm).

(A) Habitus, dorsal view; (B) cephalothorax, anterior view.

Figure 29 Potamiscus fumariatus sp. n. (Holotype, male, NCU MCP 2013.0024, 24.1 × 20.1 mm).

(A) Cephalothorax, ventral view; (B) right chela, outer view.

Figure 30 Potamiscus fumariatus sp. n. (Holotype, male, NCU MCP 2013.0024, 24.1 × 20.1 mm).

(A–D) Left G1; (A) ventral view; (B) enlarged view of distal portion, ventral view; (C) dorsal view; (D) enlarged view of distal portion, dorsal view; (E) left G2, dorsal view. Scales = 1 mm.

Material examined. Holotype, male (24.1 × 20.1 mm) (NCU MCP 2013.0024), Bailu Town, Wuding County, Yunnan Province, China, coll. Liu Shao Yan, February 25, 2004.

Diagnosis. Carapace (Fig. 28A) distinctly broader than long CW 1.20 times CL, dorsal surface (Fig. 28B) relatively flat, cervical grooves indistinct, weakly developed, reaching postorbital cristae; epigastric cristae rounded, rugose, appearing almost confluent with postorbital cristae, weakly separated from postorbital cristae by indistinct groove, region between epigastric cristae and frontal margin rugose; frontal region narrow; postorbital cristae gently slanting posterolaterally toward anterolateral margin; regions behind epigastric and postorbital cristae rugose; antennular fossae (Fig. 28B) slit-like in anterior view; cleft between external orbital angle and epibranchial tooth narrow; epibranchial tooth distinct, developed, spine-like; anterolateral margin convex, cristate, very weakly serrated being almost smooth, running inward posteriorly. Exopod of third maxilliped without flagellum. Suture between sternites 3 and 4 absent (Fig. 29A); sterno-pleonal cavity reaching imaginary line joining middle of bases of cheliped coxae. Male pleon (Fig. 29A) narrowly triangular; telson broadly triangular. G1 (Figs. 30A–30D) subterminal segment sinuous, broad; terminal segment short, columnar, with broadly flattened truncate, coiled (when viewed dorsally) tip, shorter than half the length of subterminal segment, with groove for G2 marginal.

Etymology. The species name is derived from an arbitrary combination of the Latin “fumario” (=chimney) and -atus (a suffix to form adjective indicating the possession of a thing or a quality), alluding to the possession of chimney-like G1s.

Remarks. Potamiscus fumariatus sp. n. is so far known by a single specimen collected together with another species, Aparapotamon grahami (Rathbun, 1931). However, it can be easily distinguished from the latter species by the absence of the flagellum on the exopod of the third maxilliped and its distinct G1 structure.

It most closely resembles Potamiscus motuoensis Dai, 1990, in general carapace morphology and in the G1 terminal segment being columnar-shaped and being shorter than half the length of the subterminal segment. Nevertheless, Potamiscus fumariatus can be easily distinguished from Potamiscus motuoensis by the following characters: the anterolateral margin of the carapace is less convex laterally (vs. anterolateral margin of carapace more convex laterally); the epibranchial tooth is spine-like (vs. epibranchial tooth triangular and low); and the G1 terminal segment has a broadly flattened truncate tip (vs. G1 terminal segment with a blunt tapering tip) (Figs. 28A and 30A–30D vs. Figs. 19C and 19G; Dai, 1990: fig. 69-4, pl. 9 fig. 3; Dai, 1999: fig. 106 (5, 6), pl. 13 fig. 1).

Distribution. Bailu Town, Wuding County, Yunnan Province, China.

Potamiscus crassus sp. n.

urn:lsid:zoobank.org:act:AA45FEE-505F-4C92-BA19-8D8E7AD3091E

Figs. 31–34.

Figure 31 Potamiscus crassus sp. n. (Holotype, male, NCU MCP 2013.0025, 28.0 × 23.6 mm).

(A) Habitus, dorsal view; (B) cephalothorax, anterior view.

Figure 32 Potamiscus crassus sp. n. (Holotype, male, NCU MCP 2013.0025, 28.0 × 23.6 mm).

(A) Cephalothorax, ventral view; (B) left chela, outer view.

Figure 33 Potamiscus crassus sp. n. (Holotype, male, NCU MCP 2013.0025, 28.0 × 23.6 mm).

(A–D) Right G1; (A) ventral view; (B) enlarged view of distal portion, ventral view; (C) dorsal view; (D) enlarged view of distal portion, dorsal view; (E) right G2, dorsal view. Scales = 3 mm.

Figure 34 Potamiscus crassus sp. n. (Paratype, female, ZRC 2013.0562, 22.7 ×18.2 mm).

Thoracic sternum with vulvae.

Material examined. Holotype, male (28.0 × 23.6 mm) (NCU MCP 2013.0025), purchased from a market near Kunmin by Yunnan Province, China, coll. local villager, purchased by Yang Chang Man, June 10, 2002.

Paratypes: 24 males (largest 26.3 × 22.4 mm), 18 females (largest 23.3 × 18.9 mm) (ZRC 2013.0562), five males (largest 27.8 × 23.4 mm), three females (largest 25.1 × 19.8 mm) (ZRC 2013.0563), six males (largest 25.4 × 21.2 mm), four females (largest 24.2 × 19.4 mm) (RUMF-ZC-2375), same data as holotype.

Diagnosis. Carapace (Fig. 31A) broader than long, CW 1.17–1.27 times (mean 1.21, n = 7) CL dorsal surface (Fig. 31B) flat, cervical grooves indistinct, weakly developed, not reaching postorbital cristae; epigastric cristae rounded, slightly rugose, appearing almost confluent with postorbital cristae, weakly separated from postorbital cristae by indistinct groove; postorbital cristae gently slanting posterolaterally toward anterolateral margin; regions behind epigastric and postorbital cristae rugose; antennular fossae (Fig. 31B) slit-like in anterior view; orbital region broad; epibranchial tooth distinct, low, poorly developed, not triangular; anterolateral margin convex, weakly serrated, not confluent with posterolateral margin, running inward posteriorly. Epistome (Fig. 31B) posterior margin lateral parts straight, with broadly triangular, obtuse median tooth. Exopod of third maxilliped without flagellum. Suture between sternites 3 and 4 distinct forming a shelf (Fig. 31B); lateral margin of sternum 4 straight; sterno-pleonal cavity reaching imaginary line joining middle of bases of cheliped coxae. Male pleon (Fig. 32A) narrowly triangular; telson narrowly triangular, shorter than proximal width, longer than sixth somite. G1 (Figs. 33A–33D) subterminal segment strongly sinuous, broad, about 1.6 times longer than broad, with distinctive subdistal cleft along outer margin; terminal segment stocky, subconical, length about 1.5 times the proximal width, with truncate, broadly rounded tip, appearing longitudinally twisted, with groove for G2 marginal. Vulva (Fig. 34) oval, large, opening directed anterior mesioventrally, located mesial end of suture 5/6, psoterior end widely produced as short eave, opening covered with membrane.

Etymology. The specific name is derived from the Latin “crassus” (=thick, fat), alluding its very stocky G1.

Remarks. Like the previous species, Potamiscus crassus sp. n. was also found together with Aparapotamon grahami (Rathbun, 1931). However, it can be easily distinguished from the latter species by the narrowly triangular male telson, the absence of the flagellum on the exopod of the third maxilliped, and its distinct G1 structure.

Potamiscus crassus sp. n. most closely resembles Potamiscus yongshengensis Dai & Chen, 1985, in general carapace morphology and in the subconical G1 terminal segment appearing longitudinally twisted as well as possessing a truncate tip. However, Potamiscus crassus sp. n. can be easily distinguished from Potamiscus yongshengensis by the following characters: the epibranchial tooth is poorly developed and not triangular (vs. epibranchial tooth more developed and triangular); male telson is narrow, with the lateral margins being proximally narrowed abruptly (vs. male telson broadly triangular, with lateral margins of telson being only slightly concave); and the G1 terminal segment has a broadly rounded tip (vs. G1 terminal segment with a narrowly rounded tip) (Figs. 31A, 32A and 33A–33D vs. Figs. 19D and 19H; Dai & Chen, 1985: fig. 5 (4, 5, 7), pl. 1 fig. 6; Dai, 1999: fig. 102 (2, 4, 5), pl. 12 fig. 5).

Distribution. The type specimens were purchased by Mrs. Yang Chang Man at a market in the outskirts of Kunmin City.

Conclusions

The present study described eight new species of potamid freshwater crab of the genera Indochinamon Yeo & Ng, 2007, Pararanguna Dai & Chen, 1985, Parvuspotamon Dai & Bo, 1994, and Potamiscus Alcock, 1909. The present study added four species to the genus Indochinamon, and the genus currently contains 38 species (Cumberlidge & Ng, 2009; Naruse, Quynh & Yeo, 2011; Do, Nguyen & Le, 2016; present study). Nineteen out of the 38 species are distributed in China, of which 18 species are known from Yunnan. Potamiscus is also distributed in Indochina and southern China (Cumberlidge & Ng, 2009). The present study added two species, and the genus now contains 16 species, seven of which have been recorded from Yunnan. Pararanguna and Parvuspotamon were monotypic genera, but the present study described one species each for the two genera. Both genera are endemic to Yunnan Province. Yunnan was already known to host the highest number of freshwater crab species in the provinces of China, and the present study further added eight species and brings the number to 58. It is noteworthy that seven out of eight new species described in the present study were collected by local colleagues and students at our request within a relatively short period in February, 2004. Also, considering the fact that more and more species have been discovered from Yunnan as well as other provinces of China (see Introduction), it is most probable that the species diversity of this group is still understudied.

We are grateful to Peter K. L. Ng for his support and encouragement of this study, and to Darren C. J. Yeo and Ng Ngan Kee for their kind assistance and input to this manuscript. Students and staff from Chuxiong Medical College, Yunnan, and Tan Heok Hui (National University of Singapore) kindly helped collecting material for our study. We also thank Jun Chen and Shuqiang Li for their help in accessing specimens in IZAS. Thanks are also due to Chunchao Zhu and Jiexin Zou for their help during a course of this study. We appreciate valuable comments from Jose Cristopher E. Mendoza, two anonymous reviewers and Robert J Toonen, which greatly improved this paper. Baidu Map kindly approved to use the map of Yunnan Province for this paper.

Additional Information and Declarations

Competing Interests

Author Contributions

Data Availability

New Species Registration

The authors declare that they have no competing interests.

Tohru Naruse conceived and designed the experiments, performed the experiments, analyzed the data, contributed reagents/materials/analysis tools, prepared figures and/or tables, authored or reviewed drafts of the paper, approved the final draft.

Jing En Chia analyzed the data, contributed reagents/materials/analysis tools.

Xianmin Zhou authored or reviewed drafts of the paper, approved the final draft.

The following information was supplied regarding data availability:

The raw data of the material examined are included in the Methods section and each species account.

The following information was supplied regarding the registration of a newly described species:

Publication LSID:

urn:lsid:zoobank.org:pub:642C4FCB-905B-48DB-8E01-96C87249D809

Indochinamon ahkense sp. n.

urn:lsid:zoobank.org:act:98393C16-DDA8-4A13-A9A3-C8778C17E8E6

Indochinamon parpidum sp. n.

urn:lsid:zoobank.org:act:FC1B9878-292C-4492-8A8D-44C7935A6904

Indochinamon tujiense sp. n.

urn:lsid:zoobank.org:act:B1E8716A-894F-44E6-8D41-7226FAF9BE28

Indochinamon lui sp. n.

urn:lsid:zoobank.org:act:3FAF5B0F-9963-4F00-9DBD-5FDE982DD04E

Pararanguna hemicyclia sp. n.

urn:lsid:zoobank.org:act:8D4F20E5-B08C-41C1-91AC-DADE13A58725

Parvuspotamon dixuense sp. n.

urn:lsid:zoobank.org:act:1D4DBFB9-B5BB-49CB-90EA-4C862CD75859

Potamiscus fumariatus sp. n.

urn:lsid:zoobank.org:act:0A9F4D94-7238-4780-8171-9D0D6DF36881

Potamiscus crassus sp. n.

urn:lsid:zoobank.org:act:AA45FEE-505F-4C92-BA19-8D8E7AD3091E

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
