# Peer review of "Biodiversity surveys reveal eight new species of freshwater crabs (Decapoda: Brachyura: Potamidae) from Yunnan Province, China"

_PeerJ, doi:10.7717/peerj.5497_

## Round 0.1 · original submission · Major Revisions

The reviewers' comments especially those of reviewer 2 and reviewer 3 should be carefully considered when revising the manuscript.

·

Basic reporting

A straightforward description of nine new species of freshwater potamid crab from southern China. The English can be improved although the errors therein are not fatal and can be easily corrected. The proper references have been consulted and cited, the authors show a good understanding of the body of knowledge. The species are well described and are compared competently with the closest congeners.

Experimental design

A straightforward taxonomy paper, containing formal diagnoses of the new taxa and some differential comparisons with related taxa.

Validity of the findings

I agree with the authors' findings.

Additional comments

Excellent work! But please check grammar and wording, as these may affect the clarity and reception of scientific information. Many of your B/W photos have their backgrounds not uniformly blacked out, please correct these. A few comparative species need their geographic ranges and type localities included in the Remarks, biogeography being an important feature in elucidating the systematics of this group. I have made my annotations directly onto the manuscript for your reference.

Reviewer 2 ·

Basic reporting

This work reports on important zoological discoveries from an understudied part of the world, and should be published. All of the figures (photos and line drawings) are relevant, and of publication quality. However, there are several points that should be addressed.

1. The English language should be improved to ensure that an international audience can clearly understand the text. Examples of where the language could be improved include lines 32-34, 40-45, 63-65.
2. The introduction needs more detail. It is suggested that the text in lines 45-50 be improved to include a brief discussion of the number of species and range of the genera Indochinamon, Pararanguna, Parvuspotamon, and Potamiscus.
3. For each of the nine species only a diagnosis is given and the full description is missing. To be clear, the diagnosis should list only the characters unique to the taxon, and the description should cover all characters of the species, whether unique or not.
4. It would be good to see a description of the habitat and ecology of each species, especially since most of these were collected relatively recently.
5. It would also be good to see a consideration of the conservation status of this species (e.g., whether it is found in a protected area, whether there are any immediate threats to it or its habitat, and the number of localities and its range).
6. It would be good to include a map showing the distributional range of each of these species – one map would work given that most are from a single locality.
7. Indochinamon includes 21 species, 12 of which occur in China. The five new species of Indochinamon have been compared with 5 species of this genus from China (which is good). However, why did the authors not compare their material with the other 7 species of this genus known from China (and even with the other 9 species from outside China)?
8. The same questions apply to the other new species: Pararanguna and Parvuspotamon (both formerly monotypic genera); and Potamiscus (12 species, 5 of which are from China, but only two from China were compared here, and the other 7 species were not considered).
9. There needs to be a statement of why the specimens were assigned to a genus: to Indochinamon, Pararanguna, Parvuspotamon, and Potamiscus. It is not clear why they were placed in these genera rather than in any of the other Chinese potamid genera. That is, what characters does the specimen share with the type species of the genus that justify its assignment to the genus?
10. After this statement, the authors should point out the the unique characters of the specimens that separate them from all of the other described species currently assigned to the genus. This would be the most powerful taxonomic argument to justify the establishment of a new species, given the fact that no molecular data are brought to bear in this study.
11. The authors should include a statement (e.g., in the Methods section around line 59) stating that the type material and other specimens have been preserved exclusively in ethanol and would be therefore be suitable for DNA sequencing and other molecular analyses. Alternatively they can state that all of the material has been formalin-fixed before ethanol preservation.

Experimental design

Original primary research within Aims and Scope of journal.

Validity of the findings

Conclusions are well stated, linked to original research question and limited to supporting results.

Additional comments

See above.

Reviewer 3 ·

Basic reporting

General comments:
The manuscript No. 22317 described nine new species of freshwater crabs collected from Yunnan Province, China. This is an important addition to freshwater crab biodiversity in the most species-rich province Yunnan, as well as the most species-rich country, China. However, the current version of the manuscript suffers, in my view, of serious drawbacks, including deficient data and data analysis that are described in details below. Hence I will recommend a major revision be undertaken prior to acceptance for publication in the Peer J.

Experimental design

1. Detailed information of each species described in the manuscript such as longitude and latitude, altitude, habitat environment, etc should be given. Since Chinese geographic names seem to be less common for most readers, it is preferable to provide a brief map of collecting sites.
2. Paratypes: please provide the number for each specimen, one number for several specimens is confusing, such as line 107~109.
3. In the diagnosis sections of the nine species, please give more pronounced description. Besides, explicit figures should be made, including separated figures of the third maxilliped (showing distinct traits of exopod), female thoracic sternum and genital pore, sterno-abdominal cavity with G1 in situ, and left G2.
4. In the remarks sections of the nine species, please give pronounced descriptive differences when diagnosing the new species, it should be explicitly stated what characters and how are different by use of explicit figures. For several ambiguous species, such as the two species of Indochinamon, independent data from DNA barcoding and molecular phylogenetic analyses of related clades are indispensible.
5. Figures, the authors could have done a much better job in figure quality than what is presented in the manuscript. Such as Fig. 1, Fig.4, Fig18, Fig. 20, Fig.21, Fig. 22, Fig. 25, Fig. 28.
6. Names of localities and rivers: please give more pronounced authoritative names of localities and rivers, as well as a consistent name on the same location. Such as those descriptions for Indochinamon jingguense sp. n., in lines 154-155, 160, 163-164, are ambiguous for readers.

Validity of the findings

7. Several new species or their assignment are much dubious:
1) Line 99~. Indochinamon ahkense sp. n. Please provides comparative analyses among the five related species described in the manuscript, as well as among related known species by use of tables.
2) Line 150~. Indochinamon jingguense sp. n. In terms of morphological characters, it is very similar to I. chinghungense Dai et al., 1975 in G1 and G2. Due to the lack of detailed descriptions and images of the third maxilliped, female thoracic sternum and genital pore, sterno-abdominal cavity with G1 in situ of the holotype, the new species, I. jingguense sp. n. is much dubious in the present situation. In my view, it probably is I. chinghungense Dai et al., 1975. The locality of the holotype of I. jingguense sp. n. is geographically adjacent to Jinghong (Chinghung) where I. chinghungense distributed, and these localities are all located in lower reaches of the Lancang River. Furthermore, the locality of the holotype is quite inexact, since it was sampled from a market in Simao district of Pu’er City, which is adjacent to Jinghong City.
3) Line 210~. Indochinamon parpidum sp. n. The new species is dubious.
In my view, it is probably I. hispidum (Wood-Mason, 1871). There are three subspecies, I. h. hispidum (Wood-Mason, 1871), I. h. jianchuanense Dai et Chen, 1985 and I. h. boshanense Dai et Chen, 1985. They are subsequently erected to species level. Please give comprehensive comparative analyses between the candidate new species, I. parpidum sp. n. and these three related species.
4) Line 258~. Indochinamon aozaoense sp. n. The new species is dubious, since several distinct characters, including the third maxilliped, female thoracic sternum and genital pore, sterno-abdominal cavity with G1 in situ, and G2, are not provided in descriptions and remarks, and related figures. It cannot be distinguished from I. baoshanense (Dai et Chen, 1985).
5) Line 316~. Indochinamon lui sp. n. The new species is dubious, the current descriptions, remarks and figures in the manuscript are not comparable with those for most related known species. It is similar with the former species, however, comparative analyses between these related species are not shown.
6) Line 375~. Pararanguna hemicyclius sp. n. This is undoubtedly a new species. However, current descriptions, remarks and figures in the manuscript are not comparable with those for the known related species, P. semilunatum Dai et Chen, 1985.
7) Line 438~. Parvuspotamon diyuense sp. n. The assignment of this new species is much dubious. The first is the distinct characters, including the shape of genital pore and its position on female thoracic sternum, are insufficient in remarks of the ms. The second is the length of terminal segment (distal segment) of G1is not longer than 1/2 of subternimal segment (subdistal segment), in my view, from Figs. 23 of the ms. It is not consistent to those for the genus Parvuspotamon Dai et Bo, 1994.

8) Line 492~. Potamiscus fumariatus sp. n. and Line 529~. Potamiscus crassus sp. n.: They are undoubtedly two new species. However, current descriptions, remarks and figures in the manuscript are not comparable with those for the known related species, Potamiscus Alcock, 1909. Meanwhile, the locality are dubious. In addition, the genus Potamiscus is not monophyletic, as previously reviewed by Yeo et al. Therefore, the assignment of these two species are dubious.

Additional comments

Minor concerns,
1. Line 43-44, Please recheck the name of rivers, use the unified naming of rivers in Chinese maps. And a rearrangement of these names with perspicuous westward or eastward is more proper.
2. Line 46, there are 16 genera of freshwater crabs in Yunnan, not 15 genera.
3. Line 160, the location in the sentence is quite confusing. At present, Simao is a district of Pu’er City, Yunnan Province.
4. Figure 10. Please provide left G2.
5. Figure 13. Please provide left G2.
6. Figure 16. Please provide left G2.

Annotated reviews are not available for download in order to protect the identity of reviewers who chose to remain anonymous.

---

## Round 0.2 · Minor Revisions

Although the manuscript is acceptable based on its technical merits, it does not adhere to the current PeerJ requirements for biological significance (at https://peerj.com/about/policies-and-procedures/#discipline-standards).

I have consulted with the Section Editors covering this area of the journal and they feel that the manuscript adequately reports descriptions of several new species, is scientifically sound and generally of interest. But they feel that the current level of reporting does not meet the fundamental criterion of including some overarching biological question. However, they see no reason that the manuscript would need more than a minor revision to incorporate a biological question within the current manuscript to become acceptable.

Therefore if you are able to revise the manuscript to address a biological question or hypothesis (no matter how minor) then that would make it acceptable.

Reviewer 3 ·

Basic reporting

no comment,

Experimental design

no comment,

Validity of the findings

no comment,

Additional comments

The names of rivers in figure 1 are not consistent, some of them are named ~ River, but others are named Jiang. Please check carefully.

---

## Round 0.3 · accepted · Accept

The revised manuscript is acceptable.

# #